# Emerging Insights into Molecular Mechanisms of Inflammation in Myelodysplastic Syndromes

**DOI:** 10.3390/biomedicines11102613

**Published:** 2023-09-23

**Authors:** Veronica Vallelonga, Francesco Gandolfi, Francesca Ficara, Matteo Giovanni Della Porta, Serena Ghisletti

**Affiliations:** 1Department of Experimental Oncology, European Institute of Oncology (IEO) IRCCS, 20139 Milan, Italy; 2Milan Unit, CNR-IRGB, 20090 Milan, Italy; 3IRCCS Humanitas Research Hospital, 20089 Milan, Italy; 4Department of Biomedical Sciences, Humanitas University, 20072 Milan, Italy

**Keywords:** inflammation, hematopoiesis, MDS, HSC, sc-RNAseq, sc-ATACseq, deconvolution

## Abstract

Inflammation impacts human hematopoiesis across physiologic and pathologic conditions, as signals derived from the bone marrow microenvironment, such as pro-inflammatory cytokines and chemokines, have been shown to alter hematopoietic stem cell (HSCs) homeostasis. Dysregulated inflammation can skew HSC fate-related decisions, leading to aberrant hematopoiesis and potentially contributing to the pathogenesis of hematological disorders such as myelodysplastic syndromes (MDS). Recently, emerging studies have used single-cell sequencing and muti-omic approaches to investigate HSC cellular heterogeneity and gene expression in normal hematopoiesis as well as in myeloid malignancies. This review summarizes recent reports mechanistically dissecting the role of inflammatory signaling and innate immune response activation due to MDS progression. Furthermore, we highlight the growing importance of using multi-omic techniques, such as single-cell profiling and deconvolution methods, to unravel MDSs’ heterogeneity. These approaches have provided valuable insights into the patterns of clonal evolution that drive MDS progression and have elucidated the impact of inflammation on the composition of the bone marrow immune microenvironment in MDS.

## 1. Introduction

Inflammation is a complex biological response that occurs in the body when it is exposed to various triggers, such as an infection, tissue damage, or autoimmune disorders [1]. While inflammation serves as an important defense mechanism to protect against harmful insults, prolonged or uncontrolled inflammation can have adverse effects on different cellular processes, including hematopoiesis [2,3].

Hematopoietic stem cells (HSCs) are the foundation of the hematopoietic system that are responsible for the continuous production of all blood cell lineages throughout life. HSCs possess the unique ability to self-renew, generating identical copies of themselves, and differentiating into different cell types, including red blood cells, white blood cells, and platelets [1]. Inflammation can significantly impact HSCs’ function and behavior, thereby influencing hematopoiesis [4,5]. Inflammatory signals derived from the bone marrow microenvironment impact hematopoietic stem and progenitor cells, leading to dysregulated hematopoiesis, as described in myelodysplastic syndromes (MDS) [6,7]. These conditions arise from the dysfunction of hematopoietic stem cells, resulting in the impaired differentiation and function of all types of blood cells [8].

The involvement of inflammation and innate immune responses in MDS pathogenesis have been recently reviewed elsewhere and are beyond the scope of this review [4,5,6,7,9]. Briefly, a plethora of inflammatory cytokines have been found to be elevated in MDS patients, contributing to dysplastic differentiation [10,11]. Specifically, the activation of inflammatory signals promotes the growth of aberrant MDS HSCs, while inhibiting healthy hematopoiesis [5].

Here, we aim to highlight recent significant advancements in understanding the mechanisms by which innate immune signaling and inflammation contribute to the development of MDS. A common theme that has emerged from recent studies is the induction of HSC expansion and altered responses in MDS HSC due to inflammation. This contributes to sustained myelopoiesis and a more competitive advantage compared to that of normal cells. Furthermore, we place special emphasis on the increasing importance of employing multi-omic approaches to comprehend immune dysregulation in MDS, with a focus on the role of single-cell profiling and deconvolution methods.

## 2. Inflammation Impact on MDS Hematopoiesis

Inflammation impacts human hematopoiesis across physiologic and pathologic conditions, such as infections, autoimmune disorders, and ageing. It is widely recognized that the activation of the immune system due to an infection or inflammation results in the activation of normal HSCs, ultimately causing to loss of their normal self-renewal capability [12]. In response to environmental stressors like infections, the released inflammatory mediators trigger the active cycling of HSCs (Figure 1). These molecules derived from the bone marrow microenvironment, such as pro-inflammatory cytokines and chemokines, have been shown to alter the fate and behavior of HSCs (reviewed in [1,2]). In this context, cytokines and chemokines have also been shown to regulate HSCs and promote disease progression in animal models [13]. Chronic exposure to inflammatory cytokines can result in persistent HSC cycling, and ultimately, to HSC loss (Figure 1). This process can contribute to bone marrow (BM) failure and may trigger preleukemic conditions or leukemia by inducing genetic and epigenetic changes in HSCs (Figure 1).

MDSs encompass a group of BM failure disorders distinguished by the presence of myeloid cell dysplasia. This is a heterogeneous group of hematological disorders characterized by ineffective hematopoiesis, resulting in the dysplastic and insufficient production of blood cells. In addition to inefficient hematopoiesis, a component of chronic inflammation has been consistently reported. Although the exact etiology of MDS remains elusive, emerging evidence suggests the significant involvement of innate immune signaling and inflammation in its pathogenesis [4,6,7]. Although MDS pathologies are heterogenous, a common characteristic appears to be dysregulated innate inflammation, which influences both the disease’s phenotype and its progression by inducing changes in both the hematopoietic and stromal components [5]. Specifically, MDSs frequently mirror chronic inflammatory conditions, and increased levels of serum cytokines have been reported in MDS patients [10,11]. As the disease advances, dysplastic HSCs progressively take over the BM niche, displacing normal HSCs in a process referred to as clonal expansion. In this section, we highlight recent reports mechanistically dissecting the role of inflammatory signaling and innate immune response activation due to MDS progression.

### 2.1. Inflammation Promotes HSC Expansion in CHIP

Since MDSs primarily affect older individuals, age-related inflammation may also play a role in MDS development in elderly patients. MDS can be preceded by a condition known as clonal hematopoiesis of indeterminate potential (CHIP), in which specific clones carrying pre-leukemic mutations are stably over-represented in the blood in the absence of a hematological disease [3]. The genes most frequently mutated in CHIP (such *asDnmt3a*, *Tet2*, and *Asxl1*) are also often found in other myeloid malignancies, such as MDS [14].

A common theme coming from recent studies is the association of CHIP with chronic inflammation or immune activation, such as associations between CHIP and various pro-inflammatory cytokines [15,16]. More interestingly, several mutations in genes associated with myeloid malignancies have been shown to render HSCs more susceptible to inflammation. The emerging picture is that inflammation plays a role in driving the expansion of CHIP clones [14]. In this context, HSCs’ depletion of Tet2 has been shown to display a survival advantage and proliferative phenotype in response to inflammatory challenges [17,18]. More recently, elevated HSCs proliferation following an immune challenge, such as atherosclerotic development, has been shown to drive the expansion of the *Tet2*-mutated clone [19]. On the same vein, a new mouse model carrying a recurrent *Tet2*missense mutation frequently found in individuals with CHIP was recently generated. Using sc-RNAseq profiling, it has been shown in this animal model that MDS/acute myeloid leukemia (AML) phenotype progression in aged animals correlates with an enhanced inflammatory response and the emergence of an aberrant inflammatory monocytic cell population [20]. Furthermore, the expression profile characteristic of this inflammatory population is linked to worse outcomes in AML patients, highlighting the connection between inflammation and leukemia progression (Table 1). Overall, these results indicate that leukemia development in these mice is influenced by secondary non-mutational age-related factors, as they appeared normal during their youth, but developed leukemia-like characteristics as they aged. This study also uncovered the increased activation of interferon pathways and inflammatory cytokines in the bone marrow of older mice, suggesting that an inflammatory stimulus reconfigures the transcriptional program of hematopoietic cell differentiation, accelerating myeloid leukemia development and skewing myelopoiesis toward the production of the proinflammatory monocyte type [20]. From a mechanistic point of view, the reduction of the transcription factor PU.1, a master regulator of hematopoietic stem cell and myeloid lineage, has been shown to cooperate with the Tet2 loss of function to trigger the leukemia phenotype, specifically in the setting of aging [21]. Using transposase-accessible chromatin sequencing (ATAC-seq) and bisulfite sequencing, *Tet2*mediated hypermethylated sites were identified on the PU.1 enhancer, driving the dysregulation of PU.1 expression and subsequent clonal expansion during aging [21].

On the same vein, interferon gamma (IFNγ)-signaling induced during chronic inflammation has been demonstrated to lead the proliferation of HSCs carrying loss-of-function mutations in *Dnmt3a*, the gene most commonly mutated in CHIP [22] (Table 1). In this study, a chimeric mouse model was used, where recipient mice received a transplant of 10% *Dnmt3a*−/− HSCs and 90% *Dnmt3a*+/+ HSCs. Subsequently, these mice were infected with Mycobacterium avium, which triggered an IFNγ-mediated immune response that typically depletes normal HSCs. Remarkably, within this experimental framework, there was a significant expansion of both phenotypically and functionally defined *Dnmt3a*−/− HSCs and multipotent progenitors (MPPs) only in the mice exposed to M. avium. Interestingly, a similar effect was also observed in the mice lacking one functional copy of *Dnmt3a*, resembling the situation observed in humans with *Dnmt3a*mutations [22].

Collectively, the findings from these mechanistic studies conducted on mouse models could hold significant implications for human hematopoietic disorders, as a link may be established between a history of chronic infections and an increased prevalence of clonal hematopoiesis. Further epidemiological studies are needed to address the strong correlation between inflammation markers and the growth of clones. Finally, addressing the underlying inflammation of patients is of utmost clinical importance and could potentially decelerate or prevent clonal expansion, thereby reducing the risk of cancer transformation.

### 2.2. MDS HSCs Show Altered Response to Chronic Inflammation

The innate immune system acts as the first line of defense against various pathogens and cellular stresses, and its deregulation can lead to autoimmune diseases and cancer, including MDS. Toll-like receptors (TLRs) are essential components of the innate immune system, recognizing pathogen-associated molecular patterns (PAMPs) and damage-associated molecular patterns (DAMPs). Aberrant TLR signaling has been observed in MDS patients, leading to the production of pro-inflammatory cytokines, such as tumor necrosis factor alpha (TNF-α), interleukin-1-beta (IL-1β), and interleukin-6 (IL-6), which contribute to the inflammatory microenvironment in the bone marrow [32]. Genetic studies of BM cells from MDS patients have revealed that more than half of them exhibited an increased expression of Toll-like receptors (TLRs) and other immune-related genes [33,34]. Also, TLRs downstream effectors, such as Myd88 and IRAK1 and IRAK4 kinases, are also overexpressed or constitutively activated in MDS patients [35,36]. Moreover, the aberrant expression of tumor necrosis factor receptor-associated factor 6 (TRAF6), a TLR effector with ubiquitin (Ub) ligase activity, has been observed to contribute to ineffective hematopoiesis in MDS [37]. In this context, previous studies have indicated that MDS bone marrow cells harboring del(5q) are dependent on TRAF6-mediated NF-κB signaling, an essential regulator of innate immunity and inflammation [36,38].

More recently, a mouse model with increased TRAF6 expression specifically in hematopoietic cells (Vav-Traf6-YFP) was used to mechanistically deconvolute the role of TRAF6 in MDS [23]. The RNA sequencing analysis of TRAF6-overexpressing HSCs showed that, despite the elevated TRAF6 levels, these cells did not exhibit a heightened inflammatory state compared to what was initially observed in MDS patients. These results indicate that while disrupted innate immune signaling is connected to MDS, additional factors, such as the inflammatory microenvironment, might be essential to fully mimic the human disease state.

On the other hand, when the mice with increased TRAF6 expression levels were subjected to an inflammatory stimulus like lipopolysaccharide (LPS), a component of bacterial cell walls that acts to induce chronic inflammation, there was a notable increase in TRAF6-overexpressing HSCs, leading to enhanced myeloid cell production when compared to that of the untreated mice [23] (Table 1). Clearly, these findings indicate that inflammation may work synergistically with dysregulated innate immune processes to promote the self-renewal of MDS HSCs, suggesting that inflammation might be more than just a modifying force, but rather an initiating factor in the progression of MDS. Thus, chronic inflammation from external sources exacerbates intrinsic immune dysregulation, contributing significantly to the competitive advantage of MDS. Mechanistically, MDS hematopoietic stem/progenitor cells (HSPCs) switched from canonical to noncanonical NF-kappaB signaling, a process that is dependent on TLR-TRAF6 signaling [23]. In conclusion, unlike normal HSPCs, MDS HSPCs showed an altered response to chronic inflammation, which contributed to sustained myelopoiesis and a more competitive advantage compared to that of normal cells [23].

### 2.3. Heterogeneity of Inflammatory States in MDS

NLRP3 (NOD-like receptor family, pyrin domain containing 3) is an essential component of the inflammasome, which is a multi-protein complex involved in the regulation of the immune response. In the context of MDS, it is well established that dysregulation of NLRP3 inflammasome activation plays a role in the pathogenesis and progression of the disease. The aberrant activation of NLRP3 has been associated with increased inflammation and cytokine production, potentially contributing to the altered hematopoiesis and immune dysfunction characteristics of MDS [39].

Moreover, the current research on the activation of the NLRP3 inflammasome pathway in MDS has elucidated how a self-sustaining cycle of sterile inflammation can result in progressive pyroptosis and impairment of the hematopoietic niche [40]. Specifically, increased levels of proinflammatory cytokines, reactive oxygen species, and alarmins induced the NF-kB-driven expression of key inflammasome components, such as NLRP3, PYCARD (pyrin domain and caspase recruitment domain), and caspase-1, as well as pro-interleukin-1b and pro interleukin-18 [40]. This evidence has been recently confirmed in a large cohort of MDS patients [24]. In this study, inflammatory pathways mediated by caspase-1, interleukin-1b, and interleukin-18 have been identified in low-risk MDS (LR-MDS) bone marrow patients [24]. Bulk RNA-seq experiments revealed a previously unrecognized heterogeneity of inflammation in these patients, as two different phenotypes with different levels of IL-1b were identified [24] (Table 1). These two different groups of patients were characterized by different mutational profiles, as one cluster contained 14/17 SF3B1-mutated cases, while the other cluster contained 8/8 del(5q) cases [24]. Moreover, the gene expression of sorted cell populations showed that the majority of the inflammasome-related genes, including IL-1b, were mainly expressed in the monocyte compartment [24]. This result confirms the dominant role of these cells in determining the bone marrow environment in MDS. Overall, this evidence reveals distinct inflammatory profiles in LR-MDS that might be a prerequisite for the stratification of anti-inflammatory therapies.

## 3. Multi-Omic Approaches to Dissect Immune Dysregulation in MDS

MDS and secondary AML (sAML) present unique challenges related to their complex differentiation hierarchies and similarities between malignant and normal cells in the ecosystem [41]. The cell population involved in MDS pathogenesis is mainly CD34-positive, as indicated in a plethora of molecular cytogenetics, transcriptional profiles, and xenograft animal studies. In this regard, it is worthy to note that CD34+ cells are heterogenous, and ongoing research is aimed at categorizing their subtypes through immunophenotyping, with recent studies indicating that defined CD34+ subsets based on immunophenotyping may display even greater diversity than previously anticipated [42].

Traditional bulk RNA sequencing techniques have provided valuable insights into the transcriptional profiles of different hematopoietic populations. However, they fail to capture the heterogeneity and transcriptional dynamics at the single-cell level. In recent years, experimental methods for analyzing single cells, such as by single-cell genomic sequencing and single-cell RNA-seq (scRNA-seq), has emerged as a powerful tool used to dissect cellular heterogeneity, lineage commitment, and regulatory networks during hematopoiesis.

Recent studies have indicated that precise clonal trajectories in MDS progression are made possible via single-cell genomic sequencing. In parallel, sc-RNAseq approaches have been increasingly used to address questions related to developmental hierarchies and interactions between malignant and immune cells. A milestone in this field was the use of single-cell transcriptomic and genotyping to parse heterogeneous AML ecosystems [25].

Furthermore, advancements in computational techniques used to decode the composition of distinct cell types within the tumor microenvironment have recently emerged. These innovative methods leverage bulk gene expression profiles to identify and characterize different cell populations, offering a comprehensive view of the complex cellular landscape in hematopoietic disorders.

The potential of single-cell sequencing techniques to elucidate hematopoiesis in physiological conditions and MDS at the single-cell resolution has been recently reviewed elsewhere [43]. In brief, sc-RNAseq studies have been instrumental in resolving cellular heterogeneity within a specific niche [44,45]. Specifically, the single-cell methods allowed researchers to comprehend that distinct bone marrow populations are able to influence the local microenvironment, thereby impacting the spatial organization of hematopoiesis [46,47,48]. In this section, we highlight the impact of recent studies using single-cell sequencing techniques to understand both intratumoral and interpatient heterogeneities in MDS, which have provided valuable insights in such a heterogeneous cell population in unprecedented detail. Specifically, we discuss how innovative single-cell sequencing methods have recently been used: (i) to uncover the patterns of clonal architecture and clonal evolution that drive the transformation from MDS to secondary AML; (ii) to elucidate the effect of inflammation on the composition of the BM immune microenvironment. In the end, deconvolution approaches used to infer the cellular composition of bulk-RNAseq samples are also discussed as valuable tools to unravel MDS heterogeneity.

### 3.1. Single-Cell Sequencing Approaches to Characterize Clonal Evolution of MDS to sAML

In the classical model supported by lineage-tracing studies on mice [49], human hematopoiesis is believed to follow a hierarchical sequence of stages, starting with HSCs and advancing towards a population of specialized progenitor cells. Conversely, with the advancements of the methods for cell-surface marker identification via fluorescence-activated cell sorting (FACS) and gene expression profiling in bulk and at the single-cell level, the idea of a continuum of undifferentiated hematopoietic stem and progenitor cells from which unilineage-restricted cells emerge was proposed [50]. This refined model of hematopoiesis challenges one of the concepts in the classical model by suggesting that multipotent progenitors have the capacity to make an initial commitment toward the megakaryocyte lineage at a very early stage, before subsequently transitioning to either an erythroid, myeloid, or lymphoid [42]. In this context, scRNA-seq data combined with the assays of chromatin accessibility at the single-cell level (scATAC-seq) were used to characterize the differentiation trajectories of hematopoietic cells [51]. More recently, high-throughtput sc-RNAseq has provided the comprehensive profiling of human HSPCs [52], revealing a continuum of cell fate bifurcations and a hierarchical-like structure in hematopoiesis. A breakthrough in this field came from a study that used single-cell transcriptomic and genotyping to comprehensively analyze the heterogeneity of AML [25] (Table 1).

By optimizing nanowell-based technology for high-throughput scRNA-seq and genotyping, over 30 thousand cells from AML patients and healthy donors were analyzed. These data were integrated into a machine learning classifier, which successfully distinguishing malignant from normal cells. In this way, six malignant AML cell types aligned along the hematopoietic stem cell-and-myeloid differentiation axis were identified [25]. This innovative approach linked developmental hierarchies to genotypes, assessed the characteristics and prognostic significance of primitive AML cells, and identified differentiated AML cells with immunomodulatory properties.

In the case of the clonal evolution of MDS to sAML, which is driven by the expansion of clones with leukemia-driver mutations, understanding clonal heterogeneity and mutation frequency is crucial for predicting the clinical outcomes. Recent single-cell sequencing studies of MDS and sAML bone marrow have unveiled distinct differentiation hierarchies linked to specific oncogenic drivers [26,27]. Implementing high-throughput single-cell DNA sequencing, patient-specific clonal evolution patterns, with some patients displaying linear evolution and others showing branching evolution, were identified [26] (Table 1). Furthermore, proteogenomic analysis revealed that leukemia-associated mutations are more prevalent in primitive and myeloid cells, suggesting a myeloid bias for these mutations. Investigating clonal evolution patterns also revealed that these mutations were acquired in specific orders [26]. In parallel, scRNA-seq was performed in two patients with paired MDS and sAML samples to analyze transcriptional changes associated with disease progression in primitive and mature cells. The disease progression was marked by an increase in primitive cell markers, and some markers of inflammation were upregulated in mature cells. However, transcriptional analyses showed complex and heterogeneous changes between the patients and cell types. For instance, one patient exhibited the downregulation of MHC genes in mature cells and interferon signatures in both primitive and mature cells, while the other patient displayed the upregulation of interferon signatures in mature cells [26]. Overall, these findings highlight the power of single-cell sequencing approaches in providing a deeper understanding of clonal evolution during disease progression.

On the same vein, a recent study has revealed previously unrecognized cell populations within MDS bone marrow, providing insights into the underlying mechanisms of the disease [28]. In this study, the analysis of sc-RNA seq data from HSPCs isolated from two MDS patients combined with the immunophenotypic characterization of a large cohort of MDS patients revealed that MDS HSCs display two distinct patterns, one with an increased content of common myeloid progenitors (CMP), while the other has a higher number of granulocyte-monocytic progenitors (GMP) [28] (Table 1). Using single-cell profiling, healthy patients followed differentiation trajectories (erythroid/megakaryocyte and myeloid/lymphoid) in line with the current view of hematopoiesis. In contrast, the MDS samples predominantly displayed a myeloid differentiation trajectory [28]. Furthermore, the analysis of differential gene expressions in these two MDS patients indicated that the CMP pattern MDS HSCs retained the transcriptional profile of the more immature long-term HSCs (LT-HSC), including the expression of transcription factors like PBX1, HLF, and MLLT3. On the contrary, GMP-pattern MDS HSCs were characterized by a myeloid signature, including the transcription factor CEBPa [28]. Concurrently, the analysis of bulk RNA-seq datasets from CMP pattern and GMP pattern MDS patients who had developed a resistance to hypomethylating agent (HMA)-based therapy revealed several significant findings. First of all, the CMP pattern MDS patients exhibited an increase expression of genes involved in promoting cell proliferation and survival, including the anti-apoptotic regulator B cell lymphoma 2 (BCL-2) [28]. Conversely, the genes associated with TNF signaling through NFkB and inflammatory responses were upregulated in the GMP pattern MDS patients with blast progression [28]. These discoveries could carry significant clinical implications, as CMP pattern MDS patients with upregulated BCL-2 were shown to benefit from venetoclax therapy.

### 3.2. sc-RNAseq Technology to Dissect Inflammatory Microenvironment

The inflammatory microenvironment in the bone marrow of MDS patients plays a crucial role in disease pathogenesis. The bone marrow stroma consists of various non-hematopoietic cells that support HSC functions during normal conditions and tissue repair. However, some stromal cells have been shown to directly contribute to myelodysplasia or leukemia development and can be altered during leukemia progression to promote tumor growth. In response to inflammation, stromal cells and the bone marrow microenvironment undergo alterations in their secretory profiles, promoting the survival and proliferation of MDS-initiating cells. Furthermore, the interactions between HSCs and the inflammatory bone marrow microenvironment are critical in shaping HSCs’ behavior. Inflammatory signals can alter the expression of adhesion molecules and chemokine receptors on HSCs, affecting their interactions with the bone marrow niche. The disruption of these interactions can lead to the egress of HSCs from the bone marrow and their mobilization into the bloodstream, impairing their ability to support proper hematopoiesis.

In studies characterizing the bone marrow stroma, the sc-RNAseq approach was fundamental to identifying distinct cell populations in the stem cell niche, regulating hematopoietic regeneration and potentially initiating leukemia [47]. In this study, by employing machine learning methods to identify different stromal cell types from scRNA-seq data, a detailed map of the BM stroma and its perturbation during pathological states has been generated [47]. Seventeen stromal subsets were identified, expressing distinct hematopoietic regulatory genes, including fibroblastic and osteoblastic subpopulations with different osteoblast differentiation trajectories [47]. The gene expression profiles of BM stromal cells indicated that heterogeneity exists within most cell types, and the presence of leukemia cells disrupts normal hematopoiesis. Emerging acute myeloid leukemia impaired mesenchymal osteogenic differentiation and reduced regulatory molecules necessary for normal hematopoiesis. Altogether, these data suggest that tissue stroma responds to malignant cells by disadvantaging normal parenchymal cells. Overall, the sc-RNAseq profiling approach provided a powerful means to reveal both discrete and continuous mechanisms of cellular identities and complexities within the bone marrow stroma. It also highlights how leukemia cells reshape the stromal environment to support their growth, while disadvantaging normal cells.

Interestingly, sc-RNAseq technology has recently been used to elucidate the effect of inflammation on the composition of the BM immune microenvironment and clinical outcomes in AML [29]. This novel study has uncovered the distinct remodeling of bone marrow in response to inflammation. Specifically, distinctive inflammatory patterns present in malignant cells of AML bone marrow, which correlates with the infiltration of atypical B cells, a dysfunctional B-cell subtype enriched in patients with high-level inflammation AML, were identified. By using scRNA-seq combined with cell surface markers (CITE-seq) on a large cohort of AML samples coupled with advanced computational methods, the RNA expression patterns of both leukemic and immune cells in the bone marrow microenvironment were distinguished [29] (Table 1). Interestingly, using CITE-seq, the authors identified a subset of patients expressing high levels of inflammatory genes in both leukemic and non-leukemic bone marrow cells. The inflammatory signature in the malignant cells included genes associated with class II antigen presentation, alarmins, chemokines, and interferon response pathways [29]. These findings appear to be clinically relevant, as the investigation of extensive bulk RNA-seq datasets revealed that patients exhibiting a high-level inflammation signature had worse overall survival chances [29]. These studies provide the conceptual basis for classifying patients with AML based on the immune microenvironment. It has to be established whether monitoring and regulating inflammation in patients with MDS/AML is important in determining their treatment and prognosis.

### 3.3. Deconvolution Methods to Dissect MDS Heterogeneity

Though scRNA-seq has been instrumental in understanding transcriptional dysregulation in MDS, uncovering the changes in gene expression profiles and signaling pathways, its high costs make it impractical for studying the large patient cohorts needed to effectively manage MDS heterogeneity. In parallel, bulk RNA sequencing (bulk RNA-seq) has been widely used to study gene expression profiles in heterogeneous samples. However, interpreting bulk RNA-seq data can be challenging due to the presence of multiple cell types, each with distinct gene expression signatures. This cellular heterogeneity, especially in hematopoietic tissue, hinders the identification of specific gene activity in individual cell types.

To address this issue and achieve the more accurate molecular profiling of complex biological samples, the inference of relative cellular composition has emerged as a powerful tool. In recent decades, various computational techniques have been developed, employing deconvolution methods to disentangle cell mixtures and calculate the relative proportions of the different cellular components (Figure 2).

These methods utilize computational algorithms to estimate the proportions of different cell types within bulk RNA-seq samples by analyzing the observed gene expression patterns. In simpler terms, deconvolution algorithms break down a mixture of various cell types into their individual components and calculate their respective proportions or ratios. In some cases, these algorithms can also calculate the overall expression signal of genes in each cell types. In addition, to validate the accuracy of deconvolution predictions, experimental validation using independent techniques like flow cytometry or single-cell RNA-seq can be performed.

The fundamental principles of deconvolution methods rely on using reference expression profiles or reference datasets that represent distinct cell types (Figure 2). There are two main categories of deconvolution methods: reference-based and reference-free methods. Reference-based methods use these reference expression profiles to estimate the cell-type proportions. On the other hand, reference-free methods employ statistical and machine learning approaches to infer cell-type proportions without relying on prior knowledge of reference profiles.

Thus, the significant limitation of most deconvolution approaches is the requirement for a reference profile of cell-type expression. The choice of reference has been found to be more crucial than the methodology itself is in determining deconvolution performance [53]. However, selecting appropriate cell types for the reference is not always straightforward, and obtaining matched samples for reference generation can be challenging. To improve the robustness and generalization of deconvolutions, it has been suggested that researchers integrate multiple studies with varying experimental conditions and technical platforms [54]. In addition, both bulk deconvolution methodologies and those that use scRNA-seq data as a reference perform best when applied to data on a linear scale, and the choice of normalization can significantly impact some methods. Additionally, neglecting to include cell types present in a mixture within the reference leads to considerably poorer results, regardless of the previous choices made during the deconvolution process [53].

To date, increasingly computational methods have been developed to infer cell type proportions from bulk transcriptomics data. For example, CIBERSORT (Cell-type Identification By estimating Relative Subsets Of known RNA Transcripts) uses the support vector regression algorithm to deconvolute the bulk gene expression profiles (GEPs) into cell type compositions (CTCs) based on a reference matrix that comprises the gene expression signatures (GES) of cell types of interest [55,56]. Other approaches focusing on bulk RNA-seq include methods such as MuSiC [57], Scaden [58], DeconRNASeq [59], and SCDC [60]. Recently, also, a pre-trained, context-free, deep learning foundation model for universal cell-type deconvolution called “UniCell” was introduced [61].

In the context of MDS, deconvolution approaches are becoming increasingly valuable. This is due to the availability of a growing number of RNA-seq datasets generated from different subsets of MDS patients, making them a valuable resource for understanding MDS heterogeneity (Figure 2). In addition, robust signature matrix can be obtained via the published sc-RNAseq of CD34-positive HSCs (Figure 2). Furthermore, it is crucial to keep in mind that CD34+ cells in MDS are not a homogenous cell population (Figure 2). The proportion of each cell type varies during the progression of the disease. For instance, there has been evidence of the expansion of common myeloid progenitors (CMP) and granulocyte-monocyte progenitor (GMP) populations, while the common lymphoid progenitor (CLP) cell compartment decreases [62].

Recently, the potential significance of identifying CTCs in the bone marrow of patients to predict the outcome of AML was explored [30]. In this study, a cell-type-specific gene expression signature (GES) reference matrix was generated by analyzing AML single-cell RNA sequencing data. Then, the CIBERSORT algorithm was used to estimate CTCs from bulk gene expression profiles using this custom reference matrix. The resulting AML prognostic model showed a comparable performance to those of previous gene expression-based models and served as an independent prognostic factor for AML (Table 1) [30].

In the context of MDS, a recent study applied CIBERSORTx to assess the relative percentages of immune cells in the bone marrow of 316 patients with primary MDS [31] (Table 1). Deconvolution analysis revealed that patients with a lower percentage of unpolarized macrophages (M0), but higher infiltration of macrophages M2 and eosinophils in the bone marrow, had adverse prognoses. Furthermore, high-risk immune cell scores were associated with NF-κB signaling, oxidative stress, and leukemic stem cell signature pathways [31]. Interestingly, this study utilized the deconvolution method to create an immune cell scoring system (ICSS), which was closely linked to the clinical characteristics and mutation patterns of patients. This scoring system could predict prognoses independently from established risk stratification systems and gene mutation statuses, offering a novel complementary approach to refine risk stratification and guide future therapeutic strategies for MDS patients. This study is one of the first to utilize CIBERSORTx for analyzing the clinical significance of immune cells in the entire bone marrow in the context of MDS [31]. With these promising findings, it is expected that more studies using similar approaches on larger cohorts of MDS patients will emerge in the near future.

## 4. Discussion and Conclusions

Inflammation has emerged as a significant player in MDS pathogenesis and progression. Inflammatory signals from the bone marrow microenvironment impact hematopoietic stem and progenitor cells, disrupting normal hematopoiesis and contributing to disease progression [14]. A common theme that has emerged from recent studies is that mutations in genes associated with CHIP and myeloid malignancies, such as TET2 and DNMT3A, render HSCs more susceptible to inflammation and chronic infection [20,22]. In this scenario, the induction of HSC expansion due to inflammation contributes to sustained myelopoiesis and a more competitive advantage compared to that of normal cells [23].

scRNA-seq studies on MDS bone marrow have been instrumental in discovering previously unknown cell populations and subpopulations, providing valuable insights into the underlying mechanisms of the disease. These studies have unveiled abnormal hematopoietic stem and progenitor cells, dysplastic erythroid and myeloid lineages, as well as dysregulated immune cells, shedding light on the cellular changes driving MDS progression.

However, it is essential to keep in mind that scRNA-seq generates a static snapshot of the transcriptional landscape, which limits its ability to provide conclusive information on the dynamic changes during cell state transitions.

Single-cell RNA sequencing studies have also been valuable in understanding the role of an inflamed tumor microenvironment in myeloid malignancies, revealing distinct inflammatory cell subsets associated with a poor prognosis. Nevertheless, the complexity of the data presents computational challenges due to inherent noise and nonlinearity. Analyzing such complex data requires advanced mathematical methods beyond conventional statistics, making the mapping of complex tissues like the bone marrow a challenging task that demands expertise in both experimental and computational techniques.

Deconvolution methods applied to bulk RNA-seq data have been highly beneficial in unraveling the cellular composition and heterogeneity of hematopoietic tissue. These computational approaches allow us to extract meaningful biological insights and explore the dynamics of different cell types by identifying cell-type-specific gene expression patterns. Despite their advantages, there are still important limitations that can impact analysis, especially in the context of MDS. Among these, the similarity between different cell types can affect the accuracy of deconvolution methods, particularly when a specific cell type is present in small proportions compared to those of others. However, ongoing progress in deconvolution methods is expected to address and overcome these limitations, leading to more robust and accurate analyses. In the near future, the development of innovative deconvolution approaches has the potential to enable clinicians to extract valuable insights from the increasing number of bulk RNA-seq datasets from MDS patients. These advancements might facilitate the more precise molecular profiling of such complex and heterogenous biological samples.

## Figures and Tables

**Figure 1 biomedicines-11-02613-f001:**
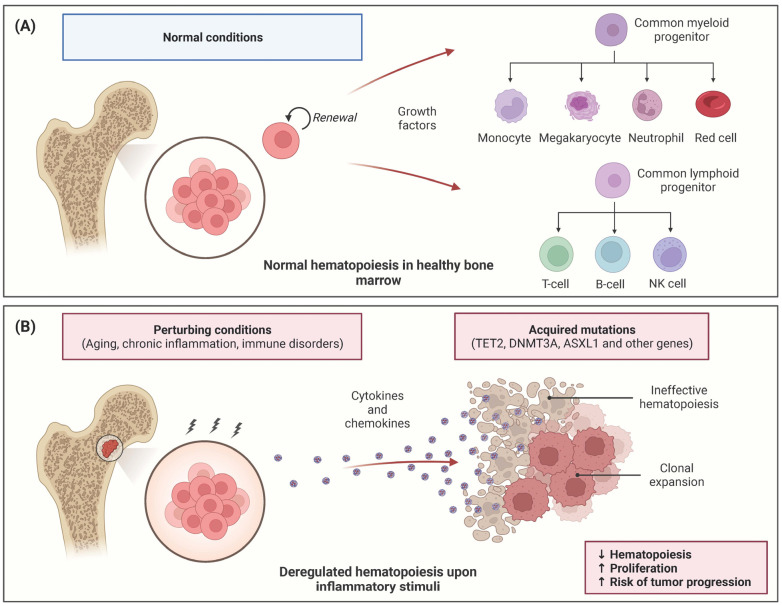
Inflammation impact on hematopoiesis (**A**,**B**). Hematopoietic stem cells (HSCs) in healthy bone marrow have the ability to self-renew and to differentiate into all blood cell lineages (**A**). Cytokines and chemokines are released from bone marrow microenvironment in response to stressors such as ageing, chronic inflammation, infections, and immune disorders (**B**). Prolonged or excessive exposure to these proinflammatory mediators can induce the loss of normal self-renewal HSCs capability and can lead to ineffective hematopoiesis. A common theme that has emerged from recent studies is that mutations in genes associated with CHIP (Clonal Hematopoiesis of Indeterminate Potential) and myeloid malignancies, such as *Tet2* and *Dnmt3a*, render HSCs more susceptible to inflammation and chronic infections. In this scenario, the induction of HSCs clonal expansion due to inflammation contributes to sustained myelopoiesis and a more competitive advantage compared to that of normal cells.

**Figure 2 biomedicines-11-02613-f002:**
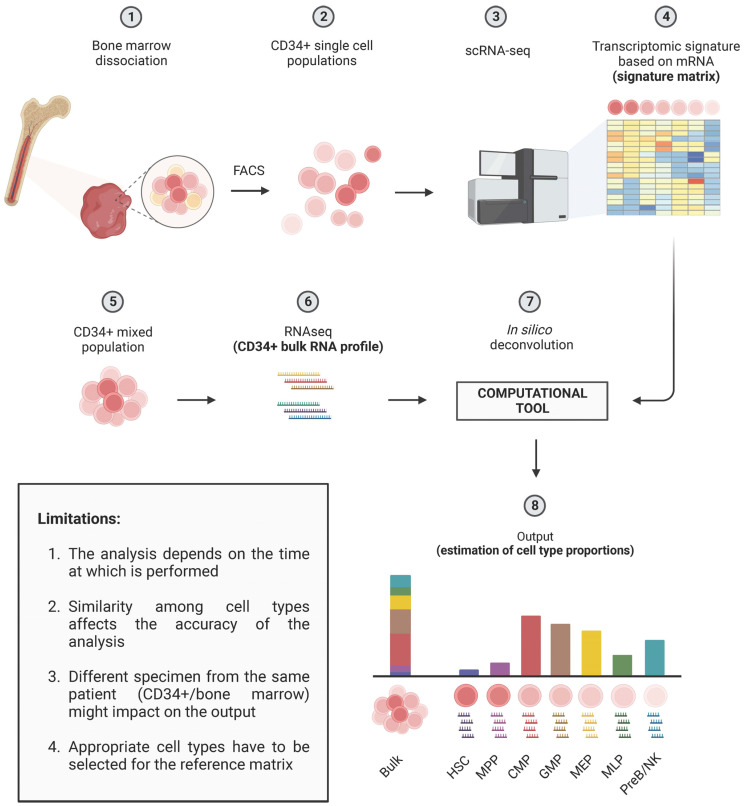
Deconvolution workflow of bulk RNA-seq dataset from CD34 positive hematopoietic stem cells. Available sc-RNAseq datasets from CD34 positive cells from human bone marrows (1–3) are used to generate a signature matrix (4). In the second step, the deconvolution methods (7) require the expression profiles of CD34-positive mixed population (5–6) and the signature matrix (4) as input bulk data to estimate the abundance of cell types in each sample (8). FACS (Fluorescence-Activated Cell Sorting), HSC (Hematopoietic Stem Cell), MPP (MultiPotent Progenitor), CMP (Common Myeloid Progenitor), GMP (Granulocyte-Monocyte Progenitor), MEP (Megakaryocyte-Erythrocyte Progenitor), MLP (Multi-Lymphoid Progenitor), and PreB/NK (Pre-B cells and Natural Killer Cells).

**Table 1 biomedicines-11-02613-t001:** List of the most relevant references discussed in the review.

Reference	Publication	Relevance
[20]	Yeaton et al., Cancer Discov (2022)	By sc-RNAseq of bone marrow from mice carrying *Tet*2 mutations, MDS/AML progression in aged animals has been shown to correlate with an enhanced inflammatory response and with the emergence of a novel population of inflammatory monocytes.
[22]	Hormaechea-Agulla et al., Cell Stem Cell (2021)	IFNγ signaling induced by chronic infection promoted the expansion of mouse *Dnmt3a*loss-of-function HSCs.
[23]	Muto et al., Nat Immunol (2020)	In a mouse model with increased TRAF6 expression in HSCs, an inflammatory stimulus has been shown to induce sustained myelopoiesis.
[24]	Schneider at al., Leukemia (2023)	Bulk RNA-seq profiling in a large cohort of MDS patients revealed distinct inflammatory pathways in subgroups of LR-MDS.
[25]	van Galen et al., Cell (2019)	sc-RNAseq and genotyping profiling of bone marrow from AML patients were used to characterize the heterogeneity of this disease.
[26]	Guess et al., Blood Cancer Discov (2022)	Single-cell sequencing of MDS and sAML bone marrow patients revealed patient-specific clonal evolution patterns.
[27]	Menssen et al., Blood Cancer Discov (2022)	Single-cell sequencing of MDS and sAML indicated that subclone expansion is a hallmark of MDS progression.
[28]	Ganan-Gomez et al., Nat Med (2022)	In this study, by sc-RNAseq of HSCs from MDS patients combined with immunophenotypic characterization, they found MDS HSCs in two distinct differentiation states.
[29]	Lasry et al., Nat Cancer (2023)	Single-cell CITE-seq profiling was used to characterize the effects of inflammation on the composition of bone marrow immune microenvironment in adult and pediatric patients with AML.
[30]	Dai et al., Front Cell Dev Biol (2021)	In this study, a novel prognostic model based on cell type composition deconvolution of bulk RNA-seq datasets of AML bone marrow or peripheral blood patients was developed.
[31]	Wang et al., Front Blood Adv (2021)	CIBERSORTx was used to estimate immune cell type compositions in bone marrow of MDS patients. A CIBERSORTx-based scoring system could predict the patients’ prognosis.

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
