# Peer review of "Emerging Insights into Molecular Mechanisms of Inflammation in Myelodysplastic Syndromes"

_biomedicines, 2023, doi:10.3390/biomedicines11102613_

Round 1

Reviewer 1 Report

The review is quite well written. I have few suggestions:

1) Abstract. Deciphering HSC Inflammation in MDS: Insights into Molecular Mechanisms. Clarify the title and replace acronyms to make the title clearer and more appealing

2) Abstract. This review summarizes recent advances on the involvement of inflammation and innate immune signaling on complex HSC homeostasis in normal and in pathophysiological conditions, such as MDS. Furthermore, we highlight the growing importance of utilizing multi-omic approaches to comprehend immune dysregulation in MDS, emphasizing the role of single-cell profiling and deconvolution methods in understanding the heterogeneity, lineage commitment, and regulatory networks involved in MDS progression. Improve this part and add some conclusions. 

3) 1. Introduction L25-39. Inflammation is a complex biological response that occurs in the body when it is ex-posed to various triggers such as infection, tissue damage, or autoimmune disorders.  While inflammation serves as an important defense mechanism to protect against harmful  insults, prolonged or uncontrolled inflammation can have adverse effects on different cel-  lular processes, including hematopoiesis. Hematopoietic stem cells (HSCs) are the foundation of the hematopoietic system, re- sponsible for the continuous production of all blood cell lineages throughout life. HSCs  possess the unique ability to self-renew, generating identical copies of themselves, and  differentiate into different cell types, including red blood cells, white blood cells, and  platelets. Inflammation can significantly impact HSC function and behavior, thereby in-fluencing hematopoiesis. Inflammatory signals derived from the bone marrow microenvironment impact hematopoietic stem and progenitor cells, leading to dysregulated hematopoiesis, as described in myelodysplastic syndromes (MDS). These conditions arise  from the dysfunction of hematopoietic stem cells, resulting in impaired differentiation and function of all types of blood cells. Please, support these information with several references.

4) L43-44. Here, we aim to highlight recent significant advancements in understanding the mechanisms by which innate immune signaling and inflammation contribute to the development of MDS. Please, improve the description of study aim.

5) 2. Inflammation impact on MDS hematopoiesis.

3. Multiomic approaches to dissect immune dysregulation in MDS.

Add a table to summarise the most important articles on the topic. 

6) L 468-476. Deconvolution methods applied to bulk RNA-seq data have been highly beneficial  in unraveling the cellular composition and heterogeneity of hematopoietic tissue. These computational approaches allow us to extract meaningful biological insights and explore  the dynamics of different cell types by identifying cell type-specific gene expression pat- terns. Despite their advantages, there are still important limitations that can impact the  analysis, especially in the context of MDS. Among these, the similarity between different  cell types can affect the accuracy of deconvolution methods, particularly when a specific cell type is present in small proportions compared to others. However, ongoing progress  in deconvolution methods is expected to address and overcome these limitations, leading  to more robust and accurate analyses. Underline the novelty of the study and the possible clinical implications.

Author Response

We would like to thank this reviewer for his/her comments, which were constructively critical and useful to improve the manuscript.

1) Abstract. Deciphering HSC Inflammation in MDS: Insights into Molecular Mechanisms. Clarify the title and replace acronyms to make the title clearer and more appealing

We changed the title and replaced the acronyms.

2) Abstract. This review summarizes recent advances on the involvement of inflammation and innate immune signaling on complex HSC homeostasis in normal and in pathophysiological conditions, such as MDS. Furthermore, we highlight the growing importance of utilizing multi-omic approaches to comprehend immune dysregulation in MDS, emphasizing the role of single-cell profiling and deconvolution methods in understanding the heterogeneity, lineage commitment, and regulatory networks involved in MDS progression. Improve this part and add some conclusions. 

We thank the reviewer for this helpful suggestion. We improved this part and we added some conclusions.

3) 1. Introduction L25-39. Inflammation is a complex biological response that occurs in the body when it is ex-posed to various triggers such as infection, tissue damage, or autoimmune disorders.  While inflammation serves as an important defense mechanism to protect against harmful  insults, prolonged or uncontrolled inflammation can have adverse effects on different cel-  lular processes, including hematopoiesis. Hematopoietic stem cells (HSCs) are the foundation of the hematopoietic system, re- sponsible for the continuous production of all blood cell lineages throughout life. HSCs  possess the unique ability to self-renew, generating identical copies of themselves, and  differentiate into different cell types, including red blood cells, white blood cells, and  platelets. Inflammation can significantly impact HSC function and behavior, thereby in-fluencing hematopoiesis. Inflammatory signals derived from the bone marrow microenvironment impact hematopoietic stem and progenitor cells, leading to dysregulated hematopoiesis, as described in myelodysplastic syndromes (MDS). These conditions arise  from the dysfunction of hematopoietic stem cells, resulting in impaired differentiation and function of all types of blood cells. Please, support these information with several references.

We appreciate the reviewer for bringing this omission to our attention. We now have inserted references for this section.

4) L43-44. Here, we aim to highlight recent significant advancements in understanding the mechanisms by which innate immune signaling and inflammation contribute to the development of MDS. Please, improve the description of study aim.

We improved this description.

5) 2. Inflammation impact on MDS hematopoiesis.

  1. Multiomic approaches to dissect immune dysregulation in MDS.

Add a table to summarise the most important articles on the topic. 

We thank the reviewer for this helpful suggestion. A table summarizing the most important articles on the topic is now present in the review.

6) L 468-476. Deconvolution methods applied to bulk RNA-seq data have been highly beneficial  in unraveling the cellular composition and heterogeneity of hematopoietic tissue. These computational approaches allow us to extract meaningful biological insights and explore  the dynamics of different cell types by identifying cell type-specific gene expression pat- terns. Despite their advantages, there are still important limitations that can impact the  analysis, especially in the context of MDS. Among these, the similarity between different  cell types can affect the accuracy of deconvolution methods, particularly when a specific cell type is present in small proportions compared to others. However, ongoing progress  in deconvolution methods is expected to address and overcome these limitations, leading  to more robust and accurate analyses. Underline the novelty of the study and the possible clinical implications.

We thank the reviewer for this helpful suggestion and we changed the manuscript accordingly.

Reviewer 2 Report

In this review article, the authors summarized and discussed the recent advances on the involvement of inflammation and innate immune signaling on complex hematopoietic stem cell (HSCs) homeostasis in normal and in pathophysiological conditions, such as myelodysplastic syndromes (MDS).

Comments

This is an interesting review article. The manuscript is well-written. The reviewer has only some minor concerns as follows:

1.     Please replace abbreviations in titles (HSC and MDS) with full names to avoid confusion for readers.

2.     In lines 203-204, is there a difference between HSPCs and HSCs? Can a name be unified (e.g. HSCs)?

3.     In line 310, please delete the full name and keep the abbreviation (HSC).

4.     In line 402, the first letter of the “estimating” needs to be capitalized.

5.     In the legend of Figure 1, the full name for FACS can be described.

Author Response

We appreciate that this reviewer found our article interesting and well-written

  1. Please replace abbreviations in titles (HSC and MDS) with full names to avoid confusion for readers.

We change the title and we replaced the abbreviations.

  1. In lines 203-204, is there a difference between HSPCs and HSCs? Can a name be unified (e.g. HSCs)?

We thank the reviewer for pointing this out. We revised the whole manuscript indicating only HSCs.

  1. In line 310, please delete the full name and keep the abbreviation (HSC).

We deleted the full name and we kept only HSC.

  1. In line 402, the first letter of the “estimating” needs to be capitalized.

We corrected the first letter of “estimating”

  1. In the legend of Figure 1, the full name for FACS can be described.

We described the full name of FACS

Reviewer 3 Report

This manuscript deals with recent advancement on the role of the innate immune signalling and inflammation in contributing to the development of MDS. The authors point out the importance of the multi-omics approaches to analyse the immune dysregulation, focusing also on the single-cell profiling and methods to interpret the results.

The review is of interest and well organized.

I strongly suggest inserting a figure to better summarize the first part of the manuscript regarding the molecular features and findings reported of the relevance of inflammation in MDS development. This is necessary to give a clear context, in which insert the second part of the review. In this figure, the cell and non-cell components found in the MDS environment should be considered.

Also, it would be of help to remark in the figure already present the limitations of the methods proposed. Actually, inflammation is by definition a dynamic process and this can be considered also for the chronic inflammation. The analysis suggested are relative to the time at which they are performed (as indicated on lines 454-456) and I would think also from where the biological material has been taken. It is conceivable that the heterogeneity reported for neoplastic diseases could be present in the same patient, giving a different scenario in different specimens of the same patient.

English language is good.

Author Response

We would like to thank this reviewer for his/her comments, which were constructively critical and useful to improve the manuscript.

I strongly suggest inserting a figure to better summarize the first part of the manuscript regarding the molecular features and findings reported of the relevance of inflammation in MDS development. This is necessary to give a clear context, in which insert the second part of the review. In this figure, the cell and non-cell components found in the MDS environment should be considered.

We have now inserted a new Figure 1, summarizing the first part of the manuscript ( relevance of inflammation in MDS development).

Also, it would be of help to remark in the figure already present the limitations of the methods proposed. Actually, inflammation is by definition a dynamic process and this can be considered also for the chronic inflammation. The analysis suggested are relative to the time at which they are performed (as indicated on lines 454-456) and I would think also from where the biological material has been taken. It is conceivable that the heterogeneity reported for neoplastic diseases could be present in the same patient, giving a different scenario in different specimens of the same patient.

We have now inserted in Figure 2 a new box about the limitations of the methods proposed.

Reviewer 4 Report

The study demonstrates that inflammation signaling of hematopoietic stem cell (HSCs) is involved in the pathogenesis of hematological disorders such as myelodysplastic syndromes.

1. Line 41-42, the involvement of inflammation and innate immune response in MDS pathogenesis may be briefly summarized and introduced in 1. Introduction

2. The potential of single-cell sequencing techniques may be described clearly with additional references in lines 226-236 in 3. Multiomics approaches to dissect immune dysregulation in MDS.

3. Deconvolution workflow of bulk RNA-seq dataset from CD34-positive hematopoietic stem cells maybe revised to add the abbreviations in the figure 1.

4. Discussion and Conclusion need to be revised to add and cite some references on inflammation and MDS pathogenesis, and summarize briefly the content of the study.

Author Response

We would like to thank this reviewer for his/her comments, which were constructively critical and useful to improve the manuscript.

  1. Line 41-42, the involvement of inflammation and innate immune response in MDS pathogenesis may be briefly summarized and introduced in 1. Introduction

In the Introduction, we briefly summarized the involvement of inflammation in MDS pathogenesis.

2. The potential of single-cell sequencing techniques may be described clearly with additional references in lines 226-236 in 3. Multiomics approaches to dissect immune dysregulation in MDS.

We better described the potential of single-cell sequencing approaches with additional references.

3. Deconvolution workflow of bulk RNA-seq dataset from CD34-positive hematopoietic stem cells maybe revised to add the abbreviations in the figure 1.

In Figure 1, we described the full name of the abbreviations.

4. Discussion and Conclusion need to be revised to add and cite some references on inflammation and MDS pathogenesis, and summarize briefly the content of the study.

In the “Discussion and Conclusion” section, we revised the sentences discussing the role of inflammation on MDS pathogenesis adding references. In addition, we briefly summarized the study content.

Round 2

Reviewer 1 Report

No further comments

Minor changes of English language are required